# OVER-THE-AIR FEDERATED TD LEARNING

**Nicolò Dal Fabbro** [1]    **Aritra Mitra** [2]    **Robert W. Heath Jr.** [2]    **Luca Schenato** [1]    **George J. Pappas** [3]

## ABSTRACT

In recent years, federated learning has been widely studied to speed up various *supervised* learning tasks at the wireless network edge under communication constraints. However, there is a lack of theoretical understanding as to whether similar speedups in sample complexity can be achieved for cooperative reinforcement learning (RL) problems subject to communication constraints. To that end, we study a federated policy evaluation problem over wireless fading channels where, to update model parameters, a central server aggregates local temporal difference (TD) update directions from $N$ agents via analog over-the-air computation (OAC). We refer to this scheme as `OAC-FedTD` and provide a rigorous finite-time convergence analysis of its performance that accounts for linear value function approximation, Markovian sampling, and channel-induced distortions and noise. Our analysis reveals the impact of the noisy fading channels on the convergence rate and establishes a linear convergence speedup w.r.t. the number of agents. This is the first non-asymptotic analysis of a cooperative RL setting under channel effects. Moreover, our proof leads to tighter bounds on the mixing time relative to existing work in federated RL (without channel effects); as such, it can be of independent interest to federated RL.

## 1 INTRODUCTION

In recent years, there has been an increasing demand for edge intelligence, where distributed machine learning algorithms are implemented at the wireless network edge on diverse datasets (Khan et al., 2020; Shi et al., 2020a). One particularly popular instance of edge computing is the federated learning (FL) framework (McMahan et al., 2017), where agents periodically coordinate with a central server to *speed up* the process of model-training for *supervised learning tasks*, while keeping raw training data private. The resulting FL problem boils down to stochastic optimization subject to communication constraints: unpredictable and heterogeneous wireless channels (Shi et al., 2020b) that have limited bandwidth (Reisizadeh et al., 2020) and suffer from channel fading effects (Sery & Cohen, 2020). In this context, the effect of imperfect channels on the convergence of the underlying optimization procedure is a topic that has been extensively studied in FL (Zhu et al., 2020), (Yang et al., 2021). Departing from this line of work on supervised learning/optimization, in this paper, we ask: *Is it possible to achieve speedups in sample-complexity for cooperative reinforcement learning (RL) problems when communica-*

*tion takes place over noisy wireless channels?* Surprisingly, there is no theoretical understanding of this question, even for the simplest of RL problems, such as policy evaluation. We seek to bridge this gap with our work.

In a standard RL setup, an agent repeatedly interacts with an environment by playing actions based on some policy, receiving observations (rewards), and then improving the policy with the aim of maximizing long-term cumulative returns. This sequential-decision making problem is captured via a Markov Decision Process (MDP), where the agent's goal is to find the optimal policy without knowledge of the transition kernels and reward functions of the MDP. As such, decision-making in RL is based on *sequential* data in the form of observations of state transitions and rewards.

For MDPs with large state and action spaces, RL algorithms typically require many data samples to achieve a desired level of accuracy (Nair et al., 2015). In this regard, the emerging paradigm of federated reinforcement learning (FRL) (Qi et al., 2021), (Khodadadian et al., 2022) seeks to reduce sample-complexity requirements by parallelizing data collection and computation across multiple agents. In FRL, agents share local model parameters or model differentials (i.e., gradient-like update directions) with a central server while keeping their states, rewards, and actions private. However, similar to FL, achieving a *convergence speedup* w.r.t the number of agents in FRL at the wireless network edge requires frequent information exchanges between the agents and the server over a *shared wireless communication medium*. This leads to an interesting ten-

---

[1]Department of Information Engineering, University of Padova, Italy [2]Department of Electrical and Computer Engineering, North Carolina State University, USA [3]Department of Electrical and Systems Engineering, University of Pennsylvania, USA. Correspondence to: Nicolò Dal Fabbro <nicolo.dalfabbro@phd.unipd.it>.

*Proceedings of the $6^{th}$ MLSys Conference Workshop on Resource-Constrained Learning in Wireless Networks*, Miami, FL, USA, 2023. Copyright 2023 by the author(s).

sion: *more agents imply more data, and hence, the hope of a greater convergence speedup; however, more agents also imply a larger communication bottleneck.*

*In this work, we provide the first principled understanding of this tension in FRL by considering a setting in which N agents cooperate to solve a policy evaluation problem.* We focus on policy evaluation since it is at the heart of value iteration algorithms like Watkin's Q-learning algorithm (Watkins & Dayan, 1992) for finding the optimal policy. In our setup, each agent interacts with the *same* MDP, collects observations by playing the policy to be evaluated, and employs temporal difference (TD) learning (Sutton, 1988) with linear function approximation to construct a local TD(0) update direction. The agents' local update directions are then transmitted to a central server over a shared wireless channel medium for model updating. To alleviate the communication bottleneck, we consider analog over-the-air computation (OAC) that has recently been advocated to provide large-scale, bandwidth- and energy-efficient up-link communication in FL (Amiri & Gündüz, 2020a; Krouka et al., 2022). In particular, OAC exploits the waveform-superposition property of the wireless multiple access channel (MAC) to enable the receiver (server) to obtain the average of the analog signals transmitted by the agents over the same time-frequency block (Cao et al., 2021). Compared to standard digital transmission, OAC comes with notable gains in up-link bandwidth efficiency. Furthermore, OAC has intrinsic privacy-preserving features (Amiri & Gündüz, 2020b; Sery & Cohen, 2020). However, analog signals transmitted over the air are subject to fading channel distortion and additive noise at the receiver (Sery & Cohen, 2020; Yang et al., 2021; Zhu et al., 2020). This leads to our main investigation question: *Is it still possible to achieve collaborative performance gains for federated policy evaluation under such channel distortion and noise?*

In this work, we provide an answer in the affirmative. Compared to the standard federated optimization setting, providing finite-time rates for our problem is significantly more challenging since (i) policy evaluation is *not a static optimization problem*; and (ii) the data samples are temporally correlated (since they are part of a Markov chain). In fact, even for the single-agent case, finite-time rates under Markovian sampling have only recently been established (Bhandari et al., 2018; Chen et al., 2019; Srikant & Ying, 2019). Moreover, almost all the works on multi-agent TD learning make a restrictive i.i.d. sampling assumption (Doan et al., 2019; Liu & Olshevsky, 2021). In light of the above discussion, our specific contributions are as follows.

**Contributions.** First, we formulate and study federated policy evaluation under the analog OAC model - a setting that we refer to as `OAC-FedTD`. Ours is the first work to formally study the convergence behavior of a cooperative

RL algorithm subject to the channel distortions and noise introduced by OAC.

Second, our main technical contribution is to provide a comprehensive non-asymptotic convergence analysis of `OAC-FedTD` that simultaneously accounts for Markovian sampling, function approximation, and channel effects. In Theorem 1, we prove that the sample-complexity bounds for `OAC-FedTD` exhibit an $N$-fold linear speedup relative to the vanilla single-agent TD algorithm. In particular, our bounds reveal that increasing the number of agents helps "drown out" the effect of the channel noise. Since RL algorithms are typically data-hungry, our linear speedup result is of considerable theoretical and practical significance.

Third, our proof of Theorem 1 is of independent interest to FRL. The only other paper in FRL that establishes a linear speedup under Markovian sampling is the very recent work (Khodadadian et al., 2022). However, unlike our analysis, this work does not need to contend with the additional randomness introduced by channel effects. Moreover, compared to the proof in (Khodadadian et al., 2022) that is based on Moreau envelopes for general stochastic approximation, we provide an alternate analysis that is much more transparent and *leads to sharper bounds even in the absence of channel effects*. Specifically, consistent with the single-agent case, our variance bounds bear a linear dependence on the mixing time of the underlying Markov chain. In contrast, the dependence in (Khodadadian et al., 2022) is quadratic.

## 2 SYSTEM MODEL AND `OAC-FEDTD`

We consider a setting involving $N$ agents, where all agents interact with the *same* Markov Decision Process (MDP). Let us denote the shared MDP by $\mathcal{M} = (\mathcal{S}, \mathcal{A}, \mathcal{P}, \mathcal{R}, \gamma)$, where $\mathcal{S}$ is a finite state space of size $n$, $\mathcal{A}$ is a finite action space, $\mathcal{P}$ is a set of action-dependent Markov transition kernels, $\mathcal{R}$ is a reward function, and $\gamma \in (0, 1)$ is the discount factor. We are interested in a *policy evaluation* problem where the agents exchange information via a central entity (server) to evaluate the value function associated with a policy $\mu : \mathcal{S} \rightarrow \mathcal{A}$. In what follows, we first briefly review some key concepts relevant to policy evaluation with function approximation. Then, we formally describe our communication model, objectives, and technical challenges.

**Policy Evaluation with Linear Function Approximation.** The policy $\mu$ to be evaluated induces a Markov Reward Process (MRP) with transition matrix $\mathbf{P}_\mu$ and reward function $R_\mu : \mathcal{S} \rightarrow \mathbb{R}$. The purpose of policy evaluation is to evaluate the value function $\boldsymbol{V}_\mu(s)$ for each $s \in \mathcal{S}$, where $\boldsymbol{V}_\mu(s)$ is the discounted expected cumulative reward obtained by playing policy $\mu$ starting from initial state $s$. Formally,

$$\boldsymbol{V}_\mu(s) = \mathbb{E}\left[\sum_{k=0}^{\infty} \gamma^k R_\mu(s_k)|s_0 = s\right], \quad (1)$$

where $s_k$ represents the state of the Markov chain at the discrete time-step $k$ under the action of the policy $\mu$. Our particular interest is in the RL setting where the Markov transition kernels and reward functions are *unknown*.

In several large-scale practical settings, the size $n$ of the state space $\mathcal{S}$ is large, thereby creating a major computational challenge. To work around this issue, we will resort to the popular idea of linear function approximation (Tsitsiklis & Van Roy, 1997) where $V_\mu$ is approximated by vectors in a linear subspace of $\mathbb{R}^n$ spanned by a set of $d$ basis vectors $\{\phi_\ell\}_{\ell \in [d]}$[1]; importantly, $d \ll n$. To be more precise, let us define the feature matrix $\Phi \triangleq [\phi_1, ..., \phi_d] \in \mathbb{R}^{n \times d}$. Given a weight (model) vector $\theta \in \mathbb{R}^d$, the parametric approximation $\hat{V}_\theta$ of $V_\mu$ is then given by $V(\theta) := \hat{V}_\theta = \Phi\theta$. If we denote the $s$-th row of $\Phi$ as $\phi'_s$, then the approximation of $V_\mu(s)$, in particular, is given by $\hat{V}_\theta(s) = \langle \theta, \phi'_s \rangle$. Throughout, we will make the standard assumption (Bhandari et al., 2018) that the columns of $\Phi$ are independent and that the rows are normalized, i.e., $\|\phi'_s\|_2^2 \le 1, \forall s \in \mathcal{S}$.

Given the above setup, the goal of the server-agent system is to collectively estimate the model vector $\theta^*$ corresponding to the best linear approximation of $V_\mu$ in the span of $\Phi$. To achieve this goal, we now describe a multi-agent variant of the classical TD(0) algorithm (Sutton, 1988). All agents start out from a common initial state $s_0 \in \mathcal{S}$ with an initial estimate $\theta_0 \in \mathbb{R}^m$. Subsequently, at each time-step $k \in \mathbb{N}$, a global model vector $\theta_k$ is broadcast by the server to all agents. Each agent $i \in [N]$ then takes an action $a_{i,k} = \mu(s_{i,k})$, and observes the next state $s_{i,k+1} \sim \mathbf{P}_\mu(\cdot|s_{i,k})$ and instantaneous reward $r_{i,k} = R_\mu(s_{i,k})$; here, $s_{i,k}$ is the state of agent $i$ at time-step $k$. Using the model vector $\theta_k$ and the observation tuple $o_{i,k} = (s_{i,k}, r_{i,k}, s_{i,k+1})$, agent $i$ computes the following local TD update direction:

$$\mathbf{g}_{i,k}(\theta_k, o_{i,k}) = (r_{i,k} + \gamma\langle \phi'_{s_{i,k+1}}, \theta_k \rangle - \langle \phi'_{s_{i,k}}, \theta_k \rangle)\phi'_{s_{i,k}}.$$

We will often use $\mathbf{g}_{i,k}(\theta_k)$ as a shorthand for $\mathbf{g}_{i,k}(\theta_k, o_{i,k})$. Note that although all agents play the same policy $\mu$, and interact with the same MDP, the realizations of the local observation sequences $\{o_{i,k}\}$ can differ across agents. We assume that these observation sequences are *statistically independent* across agents.[2] Intuitively, based on this independence property, one can expect that exchanging agents' local TD update directions should help reduce the variance in the estimate of $\theta^*$. In the wireless FRL framework that we consider in this paper, the exchange of local TD update directions occurs based on *over-the-air* aggregation. In what follows, we describe this scheme in detail.

**Over-the-air computation model.** We consider the typical OAC channel model that has been adopted, for example, in

---

[1]Given a positive integer $d$, we use the notation $[d] = 1, ..., d$.

[2]For each agent $i$, the observations over time are, however, correlated since they are all part of a single Markov chain.

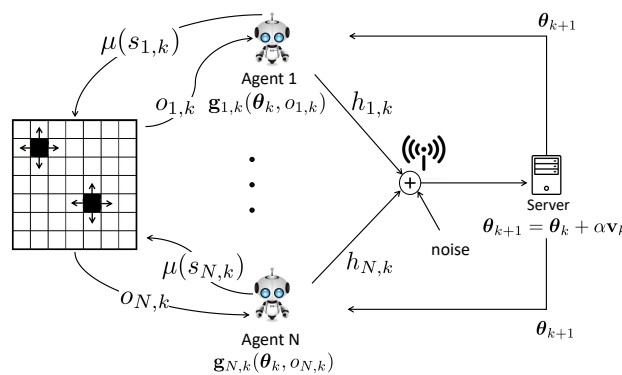

*Figure 1.* Illustration of the `OAC-FedTD` scheme. For a description of the various symbols, see Table 1.

(Amiri & Gündüz, 2020b; Cao et al., 2021; Sery & Cohen, 2020; Zhu et al., 2020). In this scheme, $N$ agents, coordinated by a central entity, synchronously transmit their local update directions as analog wireless signals. The central entity then collects the superposition of these signals; hence, the term 'over-the-air.' The analog signals are subject to fading channel distortion and to additive white Gaussian noise at the receiver.

Under the assumptions of synchronization and phase compensation (Cao et al., 2021; Sery & Cohen, 2020; Yang et al., 2021), the server at iteration $k$ obtains the following *noisy* and *distorted* global TD direction:

$$\mathbf{v}_k = \frac{1}{N}\sum_{i=1}^{N} h_{i,k}\mathbf{g}_{i,k}(\theta_k) + \mathbf{w}_k, \qquad (2)$$

where $\mathbf{w}_k \sim \mathcal{N}(0, \sigma_\mathbf{w}^2 \mathbf{I}_d)$ and $\sigma_\mathbf{w}^2 = \tilde{\sigma}_\mathbf{w}^2/N^2$, where $\tilde{\sigma}_\mathbf{w}^2$ is the additive white noise variance at the receiver. The distortion term $h_{i,k}$ is the *random* channel gain experienced by agent $i$ at iteration $k$, with mean $m_h$ and variance $\sigma_h^2$. We make the standard assumption that the random channel gain process is independent across agents and iterations. We will also assume that the random processes $\{\mathbf{w}_k\}$ and $\{h_{i,k}\}$ related to the channel effects are independent of the Markovian data tuples $\{o_{i,k}\}$. The model in (2) captures different settings of OAC. For example, the model adopted in (Cao et al., 2021) considers transmitters with adaptive power transmission. In that case, $h_{i,k} = c_{i,k}\sqrt{p_{i,k}}$, where $c_{i,k}$ is the actual channel gain, and $\sqrt{p_{i,k}}$ is the power scaling factor of device $i$ that can be adaptively adjusted to reduce the impact of the channel gain. Due to channel estimation errors (Guo et al., 2021), even in the case in which channel inversion is performed, $h_{i,k}$ is typically a random object. In general, the model considered in this paper captures any OAC framework with phase compensation, as long as the distortion $h_{i,k}$ in (2) admits first and second moments.

*Table 1.* List of notations

| Notation | Description |
|---|---|
| $o_{i,k} = (s_{i,k}, s_{i,k+1}, r_{i,k})$ | observation tuple of agent $i$ at iteration $k$ |
| $\mathbf{g}_{i,k}(\boldsymbol{\theta}_k) = \mathbf{g}(\boldsymbol{\theta}_k, o_{i,k}) \in \mathbb{R}^d$ | TD(0) update direction of agent $i$ at iteration $k$ |
| $\bar{\mathbf{g}}(\boldsymbol{\theta}_k) \in \mathbb{R}^d$ | expected TD(0) update direction at iteration $k$ |
| $\boldsymbol{\Phi} \in \mathbb{R}^{n \times d},\ \gamma \in (0,1)$ | feature matrix and discount factor |
| $\pi \in \mathbb{R}^n$ | stationary distribution of the Markov chain |
| $\mu : \mathcal{S} \to \mathcal{A}$ | policy to be evaluated by OAC-FedTD |
| $\|\cdot\|_{\mathbf{D}}$ | norm induced by $\mathbf{D} = \mathrm{diag}(\pi) \in \mathbb{R}^{n \times n}$ |
| $\omega \in \mathbb{R}$ | smallest eigenvalue of $\boldsymbol{\Sigma} = \boldsymbol{\Phi}\mathbf{D}\boldsymbol{\Phi}^\top$ |
| $h_{i,k} \in \mathbb{R}$ | channel distortion for agent $i$ at iteration $k$ |
| $\mathbf{w}_k \in \mathbb{R}^d$ | measurement noise at the receiver |
| $m_h = \mathbb{E}\left[h_{i,k}\right],\ \sigma_h^2 = \mathrm{var}(h_{i,k})$ | mean and variance of channel distortion |
| $\sigma_{\mathbf{w}}^2 \mathbf{I} = \mathrm{Var}(\mathbf{w}_k) \in \mathbb{R}^{d \times d}$ | variance of measurement noise |

Once the server receives $\mathbf{v}_k$, it updates the estimate of the parameter $\boldsymbol{\theta}_k$ according to the following update rule:

$$\boldsymbol{\theta}_{k+1} = \boldsymbol{\theta}_k + \alpha \mathbf{v}_k, \tag{3}$$

where $\alpha$ is a constant step-size/learning rate, and $\mathbf{v}_k$ is as in (2). We refer to the updating scheme described above as the over-the-air TD learning algorithm, or simply OAC-FedTD.

**Objective and Challenges.** In the rest of the paper, we aim to provide a *finite-time analysis* of OAC-FedTD. This is non-trivial for several reasons. Even in the single-agent setting, providing a non-asymptotic analysis of TD(0) without any projection step is known to be challenging due to temporal correlations between the Markov samples. To analyze OAC-FedTD, we need to deal with a multi-agent setting where two distinct sources of randomness are concurrently in place: (i) the randomness due to the time-correlated agents' trajectories, and (ii) the randomness due to the wireless fading channel. Furthermore, the final objective of OAC-FedTD is to provide a *linear convergence speedup* w.r.t. the number of agents. This requires a novel and careful analysis that we provide in the sequel.

## 3 MAIN RESULT

In this section, we state and discuss our main result pertaining to the non-asymptotic performance of OAC-FedTD. First, we need some technical preparation. We assume that the rewards are uniformly bounded, i.e., $\exists \bar{r} > 0$ such that $R_\mu(s) \le \bar{r}, \forall s \in \mathcal{S}$. This standard assumption ensures that the value function in (1) is well-defined. Next, we make another standard assumption that plays a key role in the finite-time analysis of TD algorithms (Bhandari et al., 2018; Srikant & Ying, 2019; Tsitsiklis & Van Roy, 1997).

**Assumption 1.** *The Markov chain induced by the policy $\mu$ is aperiodic and irreducible.*

As a consequence of the above assumption, the Markov

chain induced by $\mu$ admits a unique stationary distribution $\pi$ (Durrett, 2019). Let $\boldsymbol{\Sigma} = \boldsymbol{\Phi}^\top \mathbf{D}\boldsymbol{\Phi}$, where $\mathbf{D}$ is a diagonal matrix with diagonal entries given by the elements of $\pi$. Since $\boldsymbol{\Phi}$ is assumed to be full column rank, $\boldsymbol{\Sigma}$ is full rank with a strictly positive smallest eigenvalue $\omega < 1$. Next, we define the steady-state local TD direction as follows:

$$\bar{\mathbf{g}}(\boldsymbol{\theta}) \triangleq \mathbb{E}_{o_{i,k} \sim \pi}[\mathbf{g}_{i,k}(\boldsymbol{\theta}, o_{i,k})], \forall \boldsymbol{\theta} \in \mathbb{R}^d. \tag{4}$$

The *deterministic* recursion $\boldsymbol{\theta}_{k+1} = \boldsymbol{\theta}_k + \alpha\bar{\mathbf{g}}(\boldsymbol{\theta}_k)$ captures the limiting behavior of the TD(0) update rule. In (Bhandari et al., 2018), it was shown that the iterates generated by this recursion converge exponentially fast to $\boldsymbol{\theta}^*$, where $\boldsymbol{\theta}^*$ is the unique solution of the projected Bellman equation $\Pi_{\mathbf{D}}\mathcal{T}_\mu(\boldsymbol{\Phi}\boldsymbol{\theta}^*) = \boldsymbol{\Phi}\boldsymbol{\theta}^*$. Here, $\Pi_{\mathbf{D}}(\cdot)$ is the projection operator onto the subspace spanned by $\{\boldsymbol{\phi}_\ell\}_{\ell \in [d]}$ with respect to the inner product $\langle \cdot, \cdot \rangle_{\mathbf{D}}$, and $\mathcal{T}_\mu : \mathbb{R}^n \to \mathbb{R}^n$ is the policy-specific Bellman operator (Tsitsiklis & Van Roy, 1997). To analyze OAC-FedTD, we require the following definition of mixing time.

**Definition 1.** *Define* $\tau_\epsilon \triangleq \min\{t \ge 1 : \|\mathbb{E}\left[\mathbf{g}_{i,k}(\boldsymbol{\theta}, o_{i,k})|o_{i,0}\right] - \bar{\mathbf{g}}(\boldsymbol{\theta})\| \le \epsilon(\|\boldsymbol{\theta}\| + 1), \forall k \ge t, \forall \boldsymbol{\theta} \in \mathbb{R}^m, \forall i \in N, \forall o_{i,0}\}$.[3]

Assumption 1 implies that the Markov chain induced by $\mu$ mixes at a geometric rate, i.e., there exists some $K \ge 1$, such that $\tau_\epsilon \le K \log(\frac{1}{\epsilon})$. For our purpose, we will set $\epsilon = \alpha^q$, where $q$ is an integer satisfying $q \ge 2$. Unlike the centralized setting where $q = 1$ suffices (Bhandari et al., 2018; Srikant & Ying, 2019), to establish the linear speedup property, we will require $q \ge 2$. Henceforth, we will drop the subscript of $\epsilon = \alpha^q$ in $\tau_\epsilon$ and simply refer to $\tau$ as the mixing time. Let us define by $\sigma \triangleq \max\{1, \bar{r}, \|\boldsymbol{\theta}^*\|\}$ the "variance" of the observation model for our problem. Let $\delta_k^2 \triangleq \|\boldsymbol{\theta}^* - \boldsymbol{\theta}_k\|^2$, and $p_h \triangleq \max\{1, m_h^2 + \sigma_h^2\}$.

We now present the main result of our paper, which is the first finite-time result in RL with OAC. Notably, we consider the challenging case in which agents' trajectories follow a Markov process, and show that cooperation between agents provides a linear convergence speedup even under noisy analog communication over wireless fading channels.

**Theorem 1.** *Consider the update rule of OAC-FedTD in (3). There exists a universal constant $C_0 \ge 1$, such that with $\alpha \le \frac{m_h\omega(1-\gamma)}{C_0\tau p_h}$, the following holds for $T \ge 2\tau$:*

$$\mathbb{E}\left[\delta_T^2\right] \le (1 - m_h\alpha\omega(1-\gamma))^T C_1 + \frac{C_2 p_h \alpha\tau\sigma^2}{m_h\omega(1-\gamma)N}$$
$$+ \frac{C_3 p_h \tau\sigma^2\alpha^3}{m_h\omega(1-\gamma)} + \frac{C_4\alpha\tau\tilde{\sigma}_w^2 d}{m_h\omega(1-\gamma)N^2}, \tag{5}$$

*where $C_1 = 4\delta_0^2 + 2\sigma^2 + 2\frac{\tilde{\sigma}_{\mathbf{w}}^2 d}{N^2}$, and $C_2$, $C_3$, $C_4$ are universal constants.*

---

[3]Unless otherwise specified, $\|\cdot\|$ is the Euclidean norm.

A detailed proof of Theorem 1 is provided in the Appendix, where we outline the key technical challenges relative to the centralized analysis in (Srikant & Ying, 2019).

**Discussion:** We now discuss the main takeaways from Theorem 1. From (5), we first note that `OAC-FedTD` guarantees linear convergence (in the mean-square sense) to a ball around $\theta^*$ whose radius depends on the second, third and fourth terms in (5). The linear convergence rate gets slackened by both the mean distortion $m_h$, and by the choice of the step size, which needs to scale inversely with $p_h\tau$. The term $p_h$ also inflates the dominant "variance term", namely the second term in (5). So, given that $\mathbb{E}\left[h_{i,k}^2\right] = m_h^2 + \sigma_h^2$ and recalling that $p_h = \max\{1, m_h^2 + \sigma_h^2\}$, our bound clearly reveals the effect of fading distortion. This channel effect is consistent with what one observes for analogous settings in FL with OAC (Sery & Cohen, 2020). Compared with the effect of noise in FL via OAC, we note that the variance term related to the additive noise at the receiver, i.e., the fourth term in (5), gets scaled by the mixing time $\tau$. Next, compared to the centralized setting (Srikant & Ying, 2019, Theorem 7), observe that the second and fourth terms in (5) get scaled down by a factor of $N$. Moreover, the third term is $O(\alpha^3)$, i.e., it is a higher-order term that is dominated by the second term for small enough $\alpha$. *Thus, ours is the first work in MARL/FRL over wireless fading channels to establish a variance-reduction effect w.r.t. the number of agents.* With $\alpha = O(\log(NT)/T)$, we can explicitly show that each of the four terms in (5) is $O(1/NT)$, yielding the linear speedup effect we had hoped for. Finally, note that, compared to the only other very recent paper (Khodadadian et al., 2022) that establishes linear speedup under Markovian sampling (albeit, without channel effects), the second, third, and fourth terms in (5) have a tighter dependence on the mixing time $\tau$. Indeed, while we achieve a linear dependence of $O(\tau)$, which is consistent with the centralized setting (Srikant & Ying, 2019), the dependence in (Khodadadian et al., 2022, Theorem 4.1) is $O(\tau^2)$.

## 4   SIMULATION RESULTS

In this section, we provide simulation results to validate our theory. We consider an MDP with $|\mathcal{S}| = 20$ states and a feature space spanned by $d = 10$ orthonormal basis vectors; we set the discount factor to $\gamma = 0.5$ and the step size to $\alpha = 0.02$. We generate the channel distortion $h_{i,k}$ (with mean $m_h$ and variance $\sigma_h^2$) as a Rayleigh random variable, which is a widely adopted model for fading channels (Sery & Cohen, 2020). The focus of the simulations is on two aspects: (i) the effect of the OAC channel distortion on the convergence behavior and (ii) the benefit of cooperation in improving the convergence of `OAC-FedTD`, i.e., convergence speedup w.r.t. $N$. To this end, we compare the proposed `OAC-FedTD` algorithm with a vanilla version of

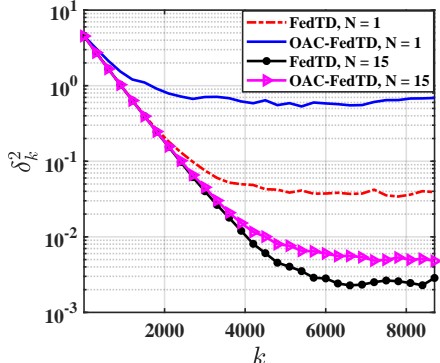

*Figure 2.* Comparison between vanilla `FedTD` and `OAC-FedTD`, in the single-agent ($N = 1$) and multi-agent ($N = 15$) case.

federated TD learning (i.e., `OAC-FedTD` without channel effects) to which we refer to as `FedTD`. For all the simulations, we set $m_h = 1$, $\sigma_h^2 \simeq 0.3$, and $\sigma_w^2 = 0.25$, which are values similar to those in (Sery & Cohen, 2020). From Fig. 2, we note that for the single-agent case ($N = 1$), the convergence error-floor of `FedTD` is smaller relative to `OAC-FedTD` with channel distortion, as expected. Crucially, by increasing the number of agents ($N = 15$), we observe that this gap can be made much smaller, thus validating the convergence speedup w.r.t. $N$ in Theorem 1.

## 5   CONCLUSION AND FUTURE WORK

We studied, for the first time, a federated policy evaluation problem where multiple agents interacting with the same MDP upload local TD(0) update directions over noisy wireless channels that introduce distortions. Our main contribution was to provide a rigorous finite-time analysis of this scheme that we referred to as `OAC-FedTD`. Our theoretical bounds reveal how the noise and distortion introduced by the channel affects sample-complexity bounds. Moreover, we showed that despite the channel effects, `OAC-FedTD` achieves a linear convergence speedup w.r.t. the number of agents under Markovian sampling. In particular, our proof of the linear speedup is novel and can be applied to other federated RL problems as well.

There are several interesting questions one can explore based on our work. For instance, the focus of this paper was solely on policy evaluation. The next natural step would be to consider the control problem and study Q learning and policy gradient algorithms over wireless channels. One can also ramp up the generality of the wireless channel model and study the effect of interference phenomenon (Yang et al., 2021). We believe that the analysis framework in our paper provides the theoretical foundation to address these interesting follow-up directions.

## Acknowledgements

This work was supported by NSF Award 1837253, and by the Italian Ministry of Education, University and Research through the PRIN project no. 2017NS9FEY.

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

## APPENDIX A

In this appendix, we will provide the detailed proof of Theorem 1. We start by introducing some definitions and preliminary results. To lighten the notation, let us define

$$
\begin{aligned}
\eta_{k,\tau}^{(i)}(\boldsymbol{\theta}) &\triangleq \|\mathbb{E}\left[\mathbf{g}_{i,k}(\boldsymbol{\theta}, o_{i,k})|o_{i,k-\tau}\right] - \bar{\mathbf{g}}(\boldsymbol{\theta})\|, \forall k \geq \tau, \forall \boldsymbol{\theta} \in \mathbb{R}^d, \forall i \in [N], \\
\delta_{k,\tau} &\triangleq \|\boldsymbol{\theta}_k - \boldsymbol{\theta}_{k-\tau}\|, \forall k \geq \tau.
\end{aligned}
\tag{6}
$$

Next, we summarize in one lemma a result from (Bhandari et al., 2018) that we will use in our analysis.

**Lemma 2.** *The following holds $\forall \boldsymbol{\theta} \in \mathbb{R}^d$:*

$$
\langle \boldsymbol{\theta}^* - \boldsymbol{\theta}, \bar{\mathbf{g}}(\boldsymbol{\theta}) \rangle \geq \omega(1-\gamma)\|\boldsymbol{\theta}^* - \boldsymbol{\theta}\|^2.
$$

We will also use the fact that the random TD update directions and their steady-state versions are 2-Lipschitz (Bhandari et al., 2018), i.e., $\forall i \in [N], \forall k \in \mathbb{N}$, and $\forall \boldsymbol{\theta}, \boldsymbol{\theta}' \in \mathbb{R}^d$, we have:

$$
\begin{aligned}
\|\bar{\mathbf{g}}(\boldsymbol{\theta}) - \bar{\mathbf{g}}(\boldsymbol{\theta}')\| &\leq 2\|\boldsymbol{\theta} - \boldsymbol{\theta}'\|, \text{ and} \\
\|\mathbf{g}_{i,k}(\boldsymbol{\theta}) - \mathbf{g}_{i,k}(\boldsymbol{\theta}')\| &\leq 2\|\boldsymbol{\theta} - \boldsymbol{\theta}'\|.
\end{aligned}
\tag{7}
$$

From (Srikant & Ying, 2019), we further have

$$
\|\mathbf{g}_{i,k}(\boldsymbol{\theta})\| \leq 2\|\boldsymbol{\theta}\| + 2\bar{r}, \forall i \in [N], \forall k \in \mathbb{N}, \forall \boldsymbol{\theta} \in \mathbb{R}^d.
\tag{8}
$$

Given that $(x+y)^2 \leq 2(x^2 + y^2), \forall x, y \in \mathbb{R}$, and the definition of $\sigma$, we will often use the following inequality:

$$
\|\mathbf{g}_{i,k}(\boldsymbol{\theta})\|^2 \leq 4(\|\boldsymbol{\theta}\| + \bar{r})^2 \leq 8(\|\boldsymbol{\theta}\|^2 + \bar{r}^2) \leq 8(\|\boldsymbol{\theta}\|^2 + \sigma^2).
\tag{9}
$$

In what follows, $\tau = \tau_\epsilon$ with $\epsilon = \alpha^2$. We now provide an intuitive outline of the proof, highlighting the challenges and the key technical steps in establishing Theorem 1.

### Outline of the Proof

The proof relies on analyzing the following recursion, which, in turn, follows directly from the update rule of OAC-FedTD:

$$
\delta_{k+1}^2 = \delta_k^2 - 2\alpha\langle \mathbf{v}_k, \boldsymbol{\theta}^* - \boldsymbol{\theta}_k \rangle + \alpha^2\|\mathbf{v}_k\|^2.
\tag{10}
$$

Let $\bar{\mathbf{g}}_N(\boldsymbol{\theta}_k) \triangleq \frac{1}{N}\sum_{i=1}^N h_{i,k}\bar{\mathbf{g}}(\boldsymbol{\theta}_k)$ and $\mathbf{g}_{h,k}(\boldsymbol{\theta}_k) \triangleq \frac{1}{N}\sum_{i=1}^N h_{i,k}\mathbf{g}_{i,k}(\boldsymbol{\theta}_k)$. Taking expectation on both sides of (10),

$$
\begin{aligned}
\mathbb{E}\left[\delta_{k+1}^2\right] = \mathbb{E}\left[\delta_k^2\right] &- 2\alpha\mathbb{E}\left[\langle \bar{\mathbf{g}}_N(\boldsymbol{\theta}_k), \boldsymbol{\theta}^* - \boldsymbol{\theta}_k \rangle\right] \\
&- 2\alpha\mathbb{E}\left[\langle \mathbf{g}_{h,k}(\boldsymbol{\theta}_k) - \bar{\mathbf{g}}_N(\boldsymbol{\theta}_k), \boldsymbol{\theta}^* - \boldsymbol{\theta}_k \rangle\right] \\
&- 2\alpha\mathbb{E}\left[\langle \mathbf{w}_k, \boldsymbol{\theta}^* - \boldsymbol{\theta}_k \rangle\right] + \alpha^2\|\mathbf{v}_k\|^2.
\end{aligned}
\tag{11}
$$

Now note that $\mathbb{E}\left[\langle \mathbf{w}_k, \boldsymbol{\theta}_k - \boldsymbol{\theta}^* \rangle\right] = \langle \mathbb{E}\left[\mathbf{w}_k\right], \mathbb{E}\left[\boldsymbol{\theta}_k - \boldsymbol{\theta}^*\right]\rangle = 0$, using the fact that the measurement noise at iteration $k$ and the iterate $\boldsymbol{\theta}_k$ are independent, and $\mathbb{E}\left[\mathbf{w}_k\right] = \mathbf{0}$. Moreover, using the fact that the distortion $h_{i,k}$ of agent $i$ at iteration $k$ and the parameter $\boldsymbol{\theta}_k$ are independent, we obtain

$$
\mathbb{E}\left[\langle \bar{\mathbf{g}}_N(\boldsymbol{\theta}_k), \boldsymbol{\theta}^* - \boldsymbol{\theta}_k \rangle\right] = \frac{1}{N}\sum_{i=1}^N \mathbb{E}\left[h_{i,k}\right]\mathbb{E}\left[\langle \bar{\mathbf{g}}(\boldsymbol{\theta}_k), \boldsymbol{\theta}^* - \boldsymbol{\theta}_k \rangle\right] = m_h\mathbb{E}\left[\langle \bar{\mathbf{g}}(\boldsymbol{\theta}_k), \boldsymbol{\theta}^* - \boldsymbol{\theta}_k \rangle\right].
\tag{12}
$$

Based on the above discussion, we can write

$$
\mathbb{E}\left[\delta_{k+1}^2\right] = \mathbb{E}\left[\delta_k^2\right] - 2\alpha m_h\langle \bar{\mathbf{g}}(\boldsymbol{\theta}_k), \boldsymbol{\theta}^* - \boldsymbol{\theta}_k \rangle + 2\alpha\mathbb{E}\left[\langle \mathbf{g}_{h,k}(\boldsymbol{\theta}_k) - \bar{\mathbf{g}}_N(\boldsymbol{\theta}_k), \boldsymbol{\theta}_k - \boldsymbol{\theta}^* \rangle\right] + \alpha^2\|\mathbf{v}_k\|^2.
\tag{13}
$$

Now define

$$
\psi_k \triangleq \langle \mathbf{g}_{h,k}(\boldsymbol{\theta}_k) - \bar{\mathbf{g}}_N(\boldsymbol{\theta}_k), \boldsymbol{\theta}_k - \boldsymbol{\theta}^* \rangle.
\tag{14}
$$

Using Lemma 2, we then obtain

$$\mathbb{E}\left[\delta_{k+1}^2\right] \leq \mathbb{E}\left[\delta_k^2\right] - 2\alpha m_h(1-\gamma)\omega\mathbb{E}\left[\delta_k^2\right] + 2\alpha\mathbb{E}\left[\psi_k\right] + \alpha^2\mathbb{E}\left[\|\mathbf{v}_k\|^2\right]. \tag{15}$$

The most challenging part of the analysis is in bounding $\mathbb{E}\left[\|\mathbf{v}_k\|^2\right]$ and $\mathbb{E}\left[\psi_k\right]$ while guaranteeing a convergence speedup w.r.t. the number of agents. In fact, even without the channel effects, this is highly non-trivial. Let us elaborate on this point. First, in standard stochastic optimization analyses, $\mathbb{E}\left[\psi_k\right]$ would vanish under the unbiasedness assumption of the stochastic gradient oracle. However, in our case, since the Markovian observations are temporally coupled, $\mathbb{E}\left[\psi_k\right]$ does not vanish. To work around this difficulty, the bounding techniques in the centralized setting, like the ones in (Bhandari et al., 2018) and (Srikant & Ying, 2019), use mixing-time arguments in conjunction with equation (9). Unfortunately, directly appealing to such techniques will fail to provide the desired convergence speedup that we seek in our multi-agent setting. The key technical step of our proof is providing a bound for $\mathbb{E}\left[\|\mathbf{v}_k\|^2\right]$ of the following form:

$$\mathbb{E}\left[\|\mathbf{v}_k\|^2\right] \leq O\left(p_h\right)\mathbb{E}\left[\delta_k^2\right] + O\left(\frac{\sigma^2 p_h}{N}\right) + O\left(\sigma^2 m_h^2\alpha^4\right) + O\left(\frac{\tilde{\sigma}_{\mathbf{w}}^2 d}{N^2}\right). \tag{16}$$

We derive this bound by appealing to the Lipschitz properties of $\mathbf{g}_{i,k}(\boldsymbol{\theta}_k)$ and performing some careful manipulations that allow us to exploit the mixing property of the Markov chain. Leveraging this key result, our next main step is to obtain a bound on $\mathbb{E}\left[\delta_{k,\tau}^2\right]$ of the following form:

$$\mathbb{E}\left[\delta_{k,\tau}^2\right] \leq O\left(\alpha^2\tau^2 p_h\right)\mathbb{E}\left[\delta_k^2\right] + O\left(\alpha^2\tau^2\frac{p_h\sigma^2}{N}\right) + O\left(\tau^2\sigma^2\alpha^4\right) + O\left(\alpha^2\tau^2\frac{\tilde{\sigma}_{\mathbf{w}}^2 d}{N^2}\right). \tag{17}$$

This result, derived in Lemma 4, turns out to play an essential role in bounding $\mathbb{E}\left[\psi_k\right]$. In particular, using Lemma 4, we show that

$$\mathbb{E}\left[\psi_k\right] \leq O\left(\alpha\tau p_h\right)\mathbb{E}\left[\delta_k^2\right] + O\left(\frac{\alpha\tau p_h\sigma^2}{N}\right) + O\left(\tau p_h\sigma^2\alpha^3\right) + O\left(\frac{\alpha\tau\tilde{\sigma}_{\mathbf{w}}^2 d}{N^2}\right).$$

This final ingredient is established in Lemma 5. Combining these bounds leads to Theorem 1. In what follows, we flesh out the above argument.

## Auxiliary Lemmas

We state and prove three lemmas that are instrumental to the proof of Theorem 1. In particular, these three results allow us to bound the terms $\mathbb{E}\left[\|\mathbf{v}_k\|^2\right]$ and $\mathbb{E}\left[\psi_k\right]$ in (15). We start by providing a bound on $\mathbb{E}\left[\|\mathbf{v}_k\|^2\right]$ of the form illustrated in (16). To that end, we state and prove the following lemma.

**Lemma 3.** *For $k \geq \tau$, we have*

$$\mathbb{E}\left[\|\mathbf{v}_k\|^2\right] \leq 8p_h\mathbb{E}\left[\delta_k^2\right] + 32\frac{\sigma^2 p_h}{N} + 8\sigma^2 m_h^2\alpha^4 + \frac{\tilde{\sigma}_{\mathbf{w}}^2 d}{N^2}. \tag{18}$$

*Proof.* Let us start by noting that the randomness in $\boldsymbol{\theta}_k$ is induced by $\{h_{i,\ell}\}_{i\in[N],\ell\in[k-1]}, \{o_{i,\ell}\}_{i\in[N],\ell\in[k-1]}$, and $\{\mathbf{w}_\ell\}_{\ell\in[k-1]}$. Based on our assumptions on the noise process, $\mathbf{w}_k$ is independent of each of these random variables and also independent of $\{h_{i,k}\}_{i\in[N]}$ and $\{o_{i,k}\}_{i\in[N]}$. Using these observations with the fact that $\mathbb{E}\left[\mathbf{w}_k\right] = \mathbf{0}$, we immediately obtain $\mathbb{E}\left[\langle\mathbf{g}_{h,k}(\boldsymbol{\theta}_k), \mathbf{w}_k\rangle\right] = \langle\mathbb{E}\left[\mathbf{g}_{h,k}(\boldsymbol{\theta}_k)\right], \mathbb{E}\left[\mathbf{w}_k\right]\rangle = 0$. This yields:

$$\mathbb{E}\left[\|\mathbf{v}_k\|^2\right] = \mathbb{E}\left[\|\mathbf{g}_{h,k}(\boldsymbol{\theta}_k)\|^2\right] + \mathbb{E}\left[\|\mathbf{w}_k\|^2\right]. \tag{19}$$

Now, note that in the centralized/single-agent TD analysis, $\|\mathbf{g}_{h,k}(\boldsymbol{\theta}_k)\|^2$ could be bounded using (8), and this would provide a term of the form $O(\delta_k^2) + O(\sigma^2)$. This approach would, however, fail to provide a linear convergence speedup with the number of agents, $N$. We will show how through a finer analysis, we can establish a tighter bound. We start by writing

$$\begin{aligned}
\|\mathbf{g}_{h,k}(\boldsymbol{\theta}_k)\|^2 &= \|\mathbf{g}_{h,k}(\boldsymbol{\theta}_k) - \mathbf{g}_{h,k}(\boldsymbol{\theta}^*) + \mathbf{g}_{h,k}(\boldsymbol{\theta}^*)\|^2 \\
&\leq \frac{2}{N^2}\left(T_1 + T_2\right),
\end{aligned} \tag{20}$$

where $T_1$ and $T_2$ are as follows:

$$T_1 = \|\sum_{i=1}^{N} h_{i,k}\mathbf{g}_{i,k}(\boldsymbol{\theta}^*)\|^2, \quad T_2 = \|\sum_{i=1}^{N} h_{i,k}(\mathbf{g}_{i,k}(\boldsymbol{\theta}_k) - \mathbf{g}_{i,k}(\boldsymbol{\theta}^*))\|^2. \tag{21}$$

We proceed to bound $T_1$ first. We express $T_1 = T_{11} + T_{12}$, with

$$T_{11} = \sum_{i=1}^{N} h_{i,k}^2 \|\mathbf{g}_{i,k}(\boldsymbol{\theta}^*)\|^2, \text{ and}$$

$$T_{12} = \sum_{\substack{i,j=1 \\ i \neq j}}^{N} h_{i,k}h_{j,k}\langle\mathbf{g}_{i,k}(\boldsymbol{\theta}^*), \mathbf{g}_{j,k}(\boldsymbol{\theta}^*)\rangle. \tag{22}$$

Using (9) and the fact that $\|\boldsymbol{\theta}^*\| \leq \sigma$, we obtain

$$\|\mathbf{g}_{i,k}(\boldsymbol{\theta}^*)\|^2 \leq 16\sigma^2, \tag{23}$$

and hence, $T_{11} \leq 16\sigma^2 \sum_{i=1}^{N} h_{i,k}^2$. Taking expectations, we thus obtain

$$\mathbb{E}[T_{11}] \leq 16\sigma^2 \mathbb{E}\left[\sum_{i=1}^{N} h_{i,k}^2\right] = 16\sigma^2 N(m_h^2 + \sigma_h^2) \leq 16N\sigma^2 p_h.$$

Next, to bound the cross-terms in $T_{12}$, we will exploit the mixing property in Definition 1. To that end, we write

$$\mathbb{E}[T_{12}] = \sum_{\substack{i,j=1 \\ i \neq j}}^{N} \mathbb{E}[h_{i,k}h_{j,k}\langle\mathbf{g}_{i,k}(\boldsymbol{\theta}^*), \mathbf{g}_{j,k}(\boldsymbol{\theta}^*)\rangle]$$

$$\overset{(a)}{=} \sum_{\substack{i,j=1 \\ i \neq j}}^{N} \mathbb{E}[h_{i,k}h_{j,k}]\,\mathbb{E}[\langle\mathbf{g}_{i,k}(\boldsymbol{\theta}^*), \mathbf{g}_{j,k}(\boldsymbol{\theta}^*)\rangle]$$

$$\overset{(b)}{=} \sum_{\substack{i,j=1 \\ i \neq j}}^{N} \mathbb{E}[h_{i,k}]\,\mathbb{E}[h_{j,k}]\langle\mathbb{E}[\mathbf{g}_{i,k}(\boldsymbol{\theta}^*)], \mathbb{E}[\mathbf{g}_{j,k}(\boldsymbol{\theta}^*)]\rangle$$

$$\overset{(c)}{=} m_h^2 \sum_{\substack{i,j=1 \\ i \neq j}}^{N} \langle\mathbb{E}[\mathbb{E}[\mathbf{g}_{i,k}(\boldsymbol{\theta}^*)|o_{i,k-\tau}] - \bar{\mathbf{g}}(\boldsymbol{\theta}^*)], \mathbb{E}[\mathbb{E}[\mathbf{g}_{j,k}(\boldsymbol{\theta}^*)|o_{j,k-\tau}] - \bar{\mathbf{g}}(\boldsymbol{\theta}^*)]\rangle$$

$$\overset{(d)}{\leq} m_h^2 \sum_{\substack{i,j=1 \\ i \neq j}}^{N} \|\mathbb{E}[\mathbb{E}[\mathbf{g}_{i,k}(\boldsymbol{\theta}^*)|o_{i,k-\tau}] - \bar{\mathbf{g}}(\boldsymbol{\theta}^*)]\|\|\mathbb{E}[\mathbb{E}[\mathbf{g}_{j,k}(\boldsymbol{\theta}^*)|o_{j,k-\tau}] - \bar{\mathbf{g}}(\boldsymbol{\theta}^*)]\|$$

$$\overset{(e)}{\leq} m_h^2 \sum_{\substack{i,j=1 \\ i \neq j}}^{N} \mathbb{E}\left[\underbrace{\|\mathbb{E}[\mathbf{g}_{i,k}(\boldsymbol{\theta}^*)|o_{i,k-\tau}] - \bar{\mathbf{g}}(\boldsymbol{\theta}^*)\|}_{\eta_{k,\tau}^{(i)}(\boldsymbol{\theta}^*)}\right]\mathbb{E}\left[\underbrace{\|\mathbb{E}[\mathbf{g}_{j,k}(\boldsymbol{\theta}^*)|o_{j,k-\tau}] - \bar{\mathbf{g}}(\boldsymbol{\theta}^*)\|}_{\eta_{k,\tau}^{(j)}(\boldsymbol{\theta}^*)}\right],$$

where (a) follows from the independence between the channel distortion gains and the Markovian tuples; (b) follows from the independence between $h_{i,k}$ and $h_{j,k}$ for $i \neq j$, and between $o_{i,k}$ and $o_{j,k}$ for $i \neq j$; (c) follows from the fact that $\bar{\mathbf{g}}(\boldsymbol{\theta}^*) = \mathbf{0}$ (Bhandari et al., 2018); (d) is a consequence of the Cauchy-Schwarz inequality; and (e) follows from Jensen's inequality. Now observe that:

$$\mathbb{E}\left[\eta_{k,\tau}^{(i)}(\boldsymbol{\theta}^*)\right] \times \mathbb{E}\left[\eta_{k,\tau}^{(j)}(\boldsymbol{\theta}^*)\right] \leq \left(\alpha^2(1 + \|\boldsymbol{\theta}^*\|)\right)^2 \leq 4\sigma^2\alpha^4. \tag{24}$$

In the step above, we used the mixing property by noting that $k \geq \tau$. We therefore obtain that $\mathbb{E}[T_{12}] \leq 4N^2 m_h^2 \sigma^2 \alpha^4$. Combining the bounds for $\mathbb{E}[T_{11}]$ and $\mathbb{E}[T_{12}]$ thus yields:

$$\mathbb{E}[T_1] \leq 16\sigma^2 N p_h + 4N^2 m_h^2 \sigma^2 \alpha^4. \tag{25}$$

Now, using (7), we see that

$$\mathbb{E}[T_2] \leq N \sum_{i=1}^{N} \mathbb{E}\left[h_{i,k}^2 \|\mathbf{g}_{i,k}(\boldsymbol{\theta}_k) - \mathbf{g}_{i,k}(\boldsymbol{\theta}^*)\|^2\right]$$

$$\leq 4N\mathbb{E}[\delta_k^2] \sum_{i=1}^{N} \mathbb{E}\left[h_{i,k}^2\right] = 4p_h N^2 \mathbb{E}[\delta_k^2]. \tag{26}$$

Combining all the bounds above, we conclude that

$$\mathbb{E}\left[\|\mathbf{g}_{h,k}(\boldsymbol{\theta}_k)\|^2\right] \leq 8p_h \mathbb{E}[\delta_k^2] + 32\frac{\sigma^2 p_h}{N} + 8\sigma^2 m_h^2 \alpha^4. \tag{27}$$

The claim of the lemma then follows from the above bound and by noting that $\mathbb{E}\left[\|\mathbf{w}_k\|^2\right] = \frac{\tilde{\sigma}_{\mathbf{w}}^2 d}{N^2}$. $\qquad\square$

Our next key result is the following.

**Lemma 4.** *Let $k \geq 2\tau$ and $\alpha \leq \frac{1}{68\tau p_h}$. We then have*

$$\mathbb{E}[\delta_{k,\tau}^2] \leq 64\alpha^2 \tau^2 p_h \mathbb{E}[\delta_k^2] + 96\alpha^2 \tau^2 \frac{p_h \sigma^2}{N} + 4\alpha^4 \tau^2 \sigma^2 + 4\alpha^2 \tau^2 \frac{\tilde{\sigma}_{\mathbf{w}}^2 d}{N^2}. \tag{28}$$

*Proof.* We start by writing

$$\delta_{k+1}^2 = \delta_k^2 - 2\alpha\langle \mathbf{v}_k, \boldsymbol{\theta}^* - \boldsymbol{\theta}_k \rangle + \alpha^2\|\mathbf{v}_k\|^2 \leq (1+\alpha)\delta_k^2 + (\alpha + \alpha^2)\|\mathbf{v}_k\|^2$$
$$\leq (1+\alpha)\delta_k^2 + 2\alpha\|\mathbf{v}_k\|^2. \tag{29}$$

Now using Lemma 3, we have

$$\mathbb{E}[\delta_{k+1}^2] \leq (1+\alpha)\mathbb{E}[\delta_k^2] + 2\alpha\left(8p_h \mathbb{E}[\delta_k^2] + 32\frac{\sigma^2 p_h}{N} + 8\sigma^2 m_h^2 \alpha^4 + \frac{\tilde{\sigma}_{\mathbf{w}}^2 d}{N^2}\right)$$

$$\leq (1+17\alpha p_h)\mathbb{E}[\delta_k^2] + \underbrace{64\alpha\frac{\sigma^2 p_h}{N} + 16\sigma^2 m_h^2 \alpha^5 + 2\alpha\frac{\tilde{\sigma}_{\mathbf{w}}^2 d}{N^2}}_{B}. \tag{30}$$

Iterating this inequality, we can obtain for any $k - \tau \leq k' \leq k$,

$$\mathbb{E}[\delta_{k'}^2] \leq (1+17\alpha p_h)^\tau \mathbb{E}[\delta_{k-\tau}^2] + B\sum_{\ell=0}^{\tau-1}(1+17\alpha p_h)^\ell. \tag{31}$$

Now using the fact that $(1+x) \leq e^x, \forall x \in \mathbb{R}$, observe that $(1+17\alpha p_h)^\ell \leq (1+17\alpha p_h)^\tau \leq e^{0.25} \leq 2$, for $\alpha \leq 1/(68p_h\tau)$. Using the same argument, we also have $\sum_{\ell=0}^{\tau-1}(1+17\alpha p_h)^\ell \leq 2\tau$. This yields:

$$\mathbb{E}[\delta_{k'}^2] \leq 2\mathbb{E}[\delta_{k-\tau}^2] + 2B\tau. \tag{32}$$

Next, note that

$$\delta_{k,\tau}^2 \leq \tau \sum_{\ell=k-\tau}^{k-1} \|\boldsymbol{\theta}_{\ell+1} - \boldsymbol{\theta}_\ell\|^2 = \tau\alpha^2 \sum_{\ell=k-\tau}^{k-1} \|\mathbf{v}_\ell\|^2. \tag{33}$$

Taking expectations on both sides of the above equation and applying Lemma 3 and (32), we get

$$
\begin{aligned}
\mathbb{E}\left[\delta_{k,\tau}^2\right] &\leq \alpha^2\tau \sum_{\ell=k-\tau}^{k-1} \left(8p_h\mathbb{E}\left[\delta_\ell^2\right] + 32\frac{\sigma^2 p_h}{N} + 8\sigma^2 m_h^2\alpha^4 + \frac{\tilde{\sigma}_{\mathbf{w}}^2 d}{N^2}\right) \\
&= 8p_h\alpha^2\tau \sum_{\ell=k-\tau}^{k-1} \mathbb{E}\left[\delta_\ell^2\right] + \frac{1}{2}\alpha\tau^2 B \\
&\leq 8p_h\alpha^2\tau \sum_{\ell=k-\tau}^{k-1} \left(2\mathbb{E}\left[\delta_{k-\tau}^2\right] + 2B\tau\right) + \frac{1}{2}\alpha\tau^2 B \\
&= 16\alpha^2\tau^2 p_h\mathbb{E}\left[\delta_{k-\tau}^2\right] + 16\alpha^2\tau^3 Bp_h + \frac{1}{2}\alpha\tau^2 B.
\end{aligned}
\tag{34}
$$

In the above steps, we used the fact that $\ell \geq \tau$ since $k \geq 2\tau$. We now proceed to simplify the resulting inequality above as follows:

$$
\begin{aligned}
\mathbb{E}\left[\delta_{k,\tau}^2\right] &\leq 16\alpha^2\tau^2 p_h\mathbb{E}\left[\delta_{k-\tau}^2\right] \\
&\quad + 16\alpha^2\tau^3 p_h\left(64\alpha\frac{\sigma^2 p_h}{N} + 16\sigma^2 m_h^2\alpha^5 + 2\alpha\frac{\tilde{\sigma}_{\mathbf{w}}^2 d}{N^2}\right) \\
&\quad + \frac{1}{2}\alpha\tau^2\left(64\alpha\frac{\sigma^2 p_h}{N} + 16\sigma^2 m_h^2\alpha^5 + 2\alpha\frac{\tilde{\sigma}_{\mathbf{w}}^2 d}{N^2}\right) \\
&= 16\alpha^2\tau^2 p_h\mathbb{E}\left[\delta_{k-\tau}^2\right] \\
&\quad + 16\alpha^3\tau^3 p_h\left(64\frac{\sigma^2 p_h}{N} + 16\sigma^2 m_h^2\alpha^4 + 2\frac{\tilde{\sigma}_{\mathbf{w}}^2 d}{N^2}\right) \\
&\quad + \frac{1}{2}\alpha^2\tau^2\left(64\frac{\sigma^2 p_h}{N} + 16\sigma^2 m_h^2\alpha^4 + 2\frac{\tilde{\sigma}_{\mathbf{w}}^2 d}{N^2}\right) \\
&\overset{(a)}{\leq} 16\alpha^2\tau^2 p_h\mathbb{E}\left[\delta_{k-\tau}^2\right] + 48\alpha^2\tau^2\frac{\sigma^2 p_h}{N} + 2\alpha^4\tau^2\sigma^2 + 2\alpha^2\tau^2\frac{\tilde{\sigma}_{\mathbf{w}}^2 d}{N^2},
\end{aligned}
\tag{35}
$$

where for (a), we used the fact that $\alpha\tau \leq \frac{1}{68 p_h}$, and that $\frac{m_h^2}{p_h} \leq 1$, implying $m_h^2\alpha \leq \frac{1}{68\tau}$. Now noting that $\delta_{k-\tau}^2 \leq 2\delta_k^2 + 2\delta_{k,\tau}^2$, we obtain

$$
\mathbb{E}\left[\delta_{k,\tau}^2\right]\left(1 - 32\alpha^2\tau^2 p_h\right) \leq 32\alpha^2\tau^2 p_h\mathbb{E}\left[\delta_k^2\right] + 48\alpha^2\tau^2\frac{\sigma^2 p_h}{N} + 2\alpha^4\tau^2\sigma^2 + 2\alpha^2\tau^2\frac{\tilde{\sigma}_{\mathbf{w}}^2 d}{N^2}.
\tag{36}
$$

Since $\alpha\tau \leq \frac{1}{68 p_h}$, we have that $1 - 32\alpha^2\tau^2 p_h \leq \frac{1}{2}$, and hence

$$
\mathbb{E}\left[\delta_{k,\tau}^2\right] \leq 64\alpha^2\tau^2 p_h\mathbb{E}\left[\delta_k^2\right] + 96\alpha^2\tau^2\frac{\sigma^2 p_h}{N} + 4\alpha^4\tau^2\sigma^2 + 4\alpha^2\tau^2\frac{\tilde{\sigma}_{\mathbf{w}}^2 d}{N^2}.
\tag{37}
$$

$\square$

Using the above lemma, we are now able to provide a bound for $\mathbb{E}\left[\psi_k\right]$, which is the last ingredient we need to prove Theorem 1.

**Lemma 5.** *Let $k \geq 2\tau$ and $\alpha \leq \frac{1}{68\tau p_h}$. We then have*

$$
\mathbb{E}\left[\psi_k\right] \leq 435\alpha\tau p_h\mathbb{E}\left[\delta_k^2\right] + 657\alpha\tau\frac{p_h\sigma^2}{N} + 30\tau p_h\sigma^2\alpha^3 + 27\alpha\tau\frac{\tilde{\sigma}_{\mathbf{w}}^2 d}{N^2}.
\tag{38}
$$

*Proof.* Recall the definition of $\bar{\mathbf{g}}_N(\boldsymbol{\theta}_k) \triangleq \frac{1}{N}\sum_{i=1}^N h_{i,k}\bar{\mathbf{g}}(\boldsymbol{\theta}_k)$ and $\mathbf{g}_{h,k}(\boldsymbol{\theta}_k) \triangleq \frac{1}{N}\sum_{i=1}^N h_{i,k}\mathbf{g}_{i,k}(\boldsymbol{\theta}_k)$. We write $\psi_k$ as $\psi_k = T_1 + T_2 + T_3 + T_4$, where

$$
\begin{aligned}
T_1 &= \langle\boldsymbol{\theta}_k - \boldsymbol{\theta}_{k-\tau}, \mathbf{g}_{h,k}(\boldsymbol{\theta}_k) - \bar{\mathbf{g}}_N(\boldsymbol{\theta}_k)\rangle, \\
T_2 &= \langle\boldsymbol{\theta}_{k-\tau} - \boldsymbol{\theta}^*, \mathbf{g}_{h,k}(\boldsymbol{\theta}_{k-\tau}) - \bar{\mathbf{g}}_N(\boldsymbol{\theta}_{k-\tau})\rangle, \\
T_3 &= \langle\boldsymbol{\theta}_{k-\tau} - \boldsymbol{\theta}^*, \mathbf{g}_{h,k}(\boldsymbol{\theta}_k) - \mathbf{g}_{h,k}(\boldsymbol{\theta}_{k-\tau})\rangle, \\
T_4 &= \langle\boldsymbol{\theta}_{k-\tau} - \boldsymbol{\theta}^*, \bar{\mathbf{g}}_N(\boldsymbol{\theta}_{k-\tau}) - \bar{\mathbf{g}}_N(\boldsymbol{\theta}_k)\rangle.
\end{aligned}
\tag{39}
$$

We now bound each of the terms $\mathbb{E}[T_1] - \mathbb{E}[T_4]$ individually. We start by observing that

$$\mathbb{E}[T_1] = \langle \boldsymbol{\theta}_k - \boldsymbol{\theta}_{k-\tau}, \mathbf{g}_{h,k}(\boldsymbol{\theta}_k) - \bar{\mathbf{g}}_N(\boldsymbol{\theta}_k) \rangle$$
$$\leq \frac{1}{2\alpha\tau} \mathbb{E}[\delta_{k,\tau}^2] + \frac{1}{2}\alpha\tau \mathbb{E}[\|\mathbf{g}_{h,k}(\boldsymbol{\theta}_k) - \bar{\mathbf{g}}_N(\boldsymbol{\theta}_k)\|^2] \qquad (40)$$
$$\leq \frac{1}{2\alpha\tau} \mathbb{E}[\delta_{k,\tau}^2] + \alpha\tau \mathbb{E}[\|\mathbf{g}_{h,k}(\boldsymbol{\theta}_k)\|^2] + \alpha\tau \mathbb{E}[\|\bar{\mathbf{g}}_N(\boldsymbol{\theta}_k) - \bar{\mathbf{g}}_N(\boldsymbol{\theta}^*)\|^2].$$

Now note that $\mathbb{E}[\|\mathbf{g}_{h,k}(\boldsymbol{\theta}_k)\|^2]$ can be bounded using the same procedure we used in (20), while for $\mathbb{E}[\delta_{k,\tau}^2]$ we can invoke Lemma 4. We also have

$$\mathbb{E}[\|\bar{\mathbf{g}}_N(\boldsymbol{\theta}_k) - \bar{\mathbf{g}}_N(\boldsymbol{\theta}^*)\|^2] \leq \frac{N}{N^2} \sum_{i=1}^{N} \mathbb{E}[h_{i,k}^2 \|\bar{\mathbf{g}}(\boldsymbol{\theta}_k) - \bar{\mathbf{g}}(\boldsymbol{\theta}^*)\|^2]$$
$$\leq \frac{4}{N} \mathbb{E}[\delta_k^2] \sum_{i=1}^{N} \mathbb{E}[h_{i,k}^2] = 4p_h \mathbb{E}[\delta_k^2]. \qquad (41)$$

Now, combining the bounds on these three terms and simplifying, we can obtain

$$\mathbb{E}[T_1] \leq 44\alpha\tau p_h \mathbb{E}[\delta_k^2] + 80\alpha\tau \frac{\sigma^2 p_h}{N} + 3\tau\sigma^2\alpha^3 + 2\alpha\tau \frac{\tilde{\sigma}_{\mathbf{w}}^2 d}{N^2}. \qquad (42)$$

We now proceed to bound $\mathbb{E}[T_3]$. We will again use the fact that $\delta_{k-\tau}^2 \leq 2\delta_k^2 + 2\delta_{k,\tau}^2$.

$$\mathbb{E}[T_3] = \mathbb{E}[\langle \boldsymbol{\theta}_{k-\tau} - \boldsymbol{\theta}^*, \mathbf{g}_{h,k}(\boldsymbol{\theta}_k) - \mathbf{g}_{h,k}(\boldsymbol{\theta}_{k-\tau}) \rangle]$$
$$= \mathbb{E}\left[\langle \boldsymbol{\theta}_{k-\tau} - \boldsymbol{\theta}^*, \frac{1}{N}\sum_{i=1}^{N} h_{i,k}(\mathbf{g}_{i,k}(\boldsymbol{\theta}_k) - \mathbf{g}_{i,k}(\boldsymbol{\theta}_{k-\tau})) \rangle\right]$$
$$= \mathbb{E}\left[\frac{1}{N}\sum_{i=1}^{N} h_{i,k}\langle \boldsymbol{\theta}_{k-\tau} - \boldsymbol{\theta}^*, \mathbf{g}_{i,k}(\boldsymbol{\theta}_k) - \mathbf{g}_{i,k}(\boldsymbol{\theta}_{k-\tau}) \rangle\right]$$
$$\leq m_h \mathbb{E}\left[\delta_{k-\tau} \frac{1}{N}\sum_{i=1}^{N} \|\mathbf{g}_{i,k}(\boldsymbol{\theta}_k) - \mathbf{g}_{i,k}(\boldsymbol{\theta}_{k-\tau})\|\right] \qquad (43)$$
$$\leq \frac{\alpha\tau}{2} m_h^2 \mathbb{E}[\delta_{k-\tau}^2] + \frac{2}{\alpha\tau} \mathbb{E}[\delta_{k,\tau}^2]$$
$$\leq \alpha\tau m_h^2 \mathbb{E}[\delta_k^2] + \alpha\tau m_h^2 \mathbb{E}[\delta_{k,\tau}^2] + \frac{2}{\alpha\tau} \mathbb{E}[\delta_{k,\tau}^2]$$
$$\leq \alpha\tau m_h^2 \mathbb{E}[\delta_k^2] + \frac{3}{\alpha\tau} \mathbb{E}[\delta_{k,\tau}^2],$$

where we have used that $\alpha\tau \leq \frac{1}{68p_h}$ and $\frac{m_h^2}{p_h} \leq 1$, which imply $m_h^2\alpha\tau \leq 1$. Applying Lemma 4, we can then get

$$\mathbb{E}[T_3] \leq \alpha\tau p_h \mathbb{E}[\delta_k^2] + \frac{3}{\alpha\tau}\left(64\alpha^2\tau^2 p_h \mathbb{E}[\delta_k^2] + 96\alpha^2\tau^2 \frac{p_h\sigma^2}{N} + 4\alpha^2\tau^2\sigma^2\alpha^2 + 4\alpha^2\tau^2 \frac{\tilde{\sigma}_{\mathbf{w}}^2 d}{N^2}\right). \qquad (44)$$

Simplifying the above bound yields:

$$\mathbb{E}[T_3] \leq 193\alpha\tau p_h \mathbb{E}[\delta_k^2] + 288\alpha\tau \frac{p_h\sigma^2}{N} + 12\tau\sigma^2\alpha^3 + 12\alpha\tau \frac{\tilde{\sigma}_{\mathbf{w}}^2 d}{N^2}. \qquad (45)$$

With analogous calculations, we can derive exactly the same bound for $\mathbb{E}[T_4]$.

We now proceed to bound $\mathbb{E}[T_2]$. For ease of notation, let us define $\mathcal{F}_{k,\tau} = (\{o_{i,k-\tau}\}_{i=1}^{N}, \boldsymbol{\theta}_{k-\tau})$. Observe:

$$\mathbb{E}[T_2] = \mathbb{E}[\mathbb{E}[T_2|\mathcal{F}_{k,\tau}]] = \mathbb{E}[\langle \boldsymbol{\theta}_{k-\tau} - \boldsymbol{\theta}^*, \frac{m_h}{N}\sum_{i=1}^{N}(\mathbb{E}[\mathbf{g}_{i,k}(\boldsymbol{\theta}_{k-\tau}, o_{i,k})|\mathcal{F}_{k,\tau}] - \bar{\mathbf{g}}(\boldsymbol{\theta}_{k-\tau}))\rangle]$$
$$\leq \mathbb{E}\left[\delta_{k-\tau} \frac{m_h}{N}\sum_{i=1}^{N} \eta_{k,\tau}^{(i)}(\boldsymbol{\theta}_{k-\tau})\right] \leq m_h\alpha^2 \mathbb{E}[\delta_{k-\tau}(1 + \|\boldsymbol{\theta}_{k-\tau}\|)].$$

Since $\alpha < 1$, we have $\delta_{k-\tau}(\delta_{k-\tau} + 2\sigma) \le \frac{\delta_{k-\tau}^2}{\alpha} + 2\sigma\delta_{k-\tau} + \alpha\sigma^2 = \left(\frac{\delta_{k-\tau}}{\sqrt{\alpha}} + \sqrt{\alpha}\sigma\right)^2 \le 2\left(\frac{\delta_{k-\tau}^2}{\alpha} + \alpha\sigma^2\right)$. Based on this observation, Lemma 4, and the fact that $m_h \le p_h$, we obtain

$$
\begin{aligned}
\mathbb{E}\left[T_2\right] &\le 2m_h\alpha^2\left(\frac{\delta_{k-\tau}^2}{\alpha} + \alpha\sigma^2\right) \\
&= 2m_h\alpha\delta_{k-\tau}^2 + 2m_h\alpha^3\sigma^2 \\
&\le 4m_h\alpha\delta_k^2 + 4m_h\alpha\delta_{k,\tau}^2 + 2m_h\alpha^3\sigma^2 \\
&\le 5p_h\alpha\tau\mathbb{E}\left[\delta_k^2\right] + \alpha\tau\frac{\sigma^2 p_h}{N} + 3\tau\sigma^2 p_h\alpha^3 + \alpha\tau\frac{\tilde{\sigma}_{\mathbf{w}}^2 d}{N^2}.
\end{aligned}
\tag{46}
$$

Combining all the terms, we can conclude the proof. $\qquad\square$

We are now in position to prove Theorem 1.

**Proof of Theorem 1**

Consider the inequality that we derived in (10). For $k \ge 2\tau$, plugging in the inequality the bounds derived in Lemma 3 and in Lemma 5, we get

$$
\begin{aligned}
\mathbb{E}\left[\delta_{k+1}^2\right] &\le \mathbb{E}\left[\delta_k^2\right] - 2\alpha m_h(1-\gamma)\omega\mathbb{E}\left[\delta_k^2\right] + 2\alpha\mathbb{E}\left[\psi_k\right] + \alpha^2\mathbb{E}\left[\|\mathbf{v}_k\|^2\right] \\
&\le \mathbb{E}\left[\delta_k^2\right] - 2\alpha m_h(1-\gamma)\omega\mathbb{E}\left[\delta_k^2\right] \\
&\quad + 2\alpha\left(435\alpha\tau p_h\mathbb{E}\left[\delta_k^2\right] + 657\alpha\tau\frac{p_h\sigma^2}{N} + 30\tau p_h\sigma^2\alpha^3 + 27\alpha\tau\frac{\tilde{\sigma}_{\mathbf{w}}^2 d}{N^2}\right) \\
&\quad + \alpha^2\left(8p_h\mathbb{E}\left[\delta_k^2\right] + 32\frac{\sigma^2 p_h}{N} + 8\sigma^2 m_h^2\alpha^4 + \frac{\tilde{\sigma}_{\mathbf{w}}^2 d}{N^2}\right) \\
&= \mathbb{E}\left[\delta_k^2\right] - 2\alpha m_h(1-\gamma)\omega\mathbb{E}\left[\delta_k^2\right] \\
&\quad + 878\alpha^2\tau p_h\mathbb{E}\left[\delta_k^2\right] + 1346\alpha^2\tau\frac{p_h\sigma^2}{N} + 61\tau p_h\sigma^2\alpha^4 + 55\alpha^2\tau\frac{\tilde{\sigma}_{\mathbf{w}}^2 d}{N^2} \\
&= \mathbb{E}\left[\delta_k^2\right] - \alpha\left(2m_h(1-\gamma)\omega - 878\alpha\tau p_h\right)\mathbb{E}\left[\delta_k^2\right] \\
&\quad + 1346\alpha^2\tau\frac{p_h\sigma^2}{N} + 61\tau p_h\sigma^2\alpha^4 + 55\alpha^2\tau\frac{\tilde{\sigma}_{\mathbf{w}}^2 d}{N^2}.
\end{aligned}
\tag{47}
$$

Hence, for $\alpha \le \frac{m_h(1-\gamma)\omega}{C_0\tau p_h}$, with $C_0 = 878$, we get

$$
\mathbb{E}\left[\delta_{k+1}^2\right] \le (1 - \alpha m_h(1-\gamma)\omega)\mathbb{E}\left[\delta_k^2\right] + 1346\alpha^2\tau\frac{p_h\sigma^2}{N} + 61\tau p_h\sigma^2\alpha^4 + 55\alpha^2\tau\frac{\tilde{\sigma}_{\mathbf{w}}^2 d}{N^2}.
\tag{48}
$$

Unrolling this inequality, we obtain

$$
\begin{aligned}
\mathbb{E}\left[\delta_T^2\right] &\le (1 - \alpha m_h(1-\gamma)\omega)^{T-2\tau}\mathbb{E}\left[\delta_{2\tau}^2\right] + C_2\frac{\alpha\tau p_h\sigma^2}{m_h(1-\gamma)\omega N} \\
&\quad + \frac{C_3\tau p_h\sigma^2\alpha^3}{m_h(1-\gamma)\omega} + \frac{C_4\alpha\tau\tilde{\sigma}_{\mathbf{w}}^2 d}{m_h(1-\gamma)\omega N^2},
\end{aligned}
\tag{49}
$$

with $C_2 = 1346$, $C_3 = 61$ and $C_4 = 55$. To conclude, we proceed to bound $\mathbb{E}\left[\delta_{2\tau}^2\right]$. Note that, for any $k \ge 0$,

$$
\mathbb{E}\left[\delta_{k+1}^2\right] \le (1+\alpha)\mathbb{E}\left[\delta_k^2\right] + 2\alpha\mathbb{E}\left[\|\mathbf{v}_k\|^2\right].
\tag{50}
$$

Observe as before (see (19)):

$$
\mathbb{E}\left[\|\mathbf{v}_k\|^2\right] = \mathbb{E}\left[\|\mathbf{g}_{h,k}(\boldsymbol{\theta}_k)\|^2\right] + \mathbb{E}\left[\|\mathbf{w}_k\|^2\right].
\tag{51}
$$

Note that $\mathbb{E}\left[\|\mathbf{w}_k\|^2\right] = \frac{\tilde{\sigma}_{\mathbf{w}}^2 d}{N^2}$ and that we can bound $\mathbb{E}\left[\|\mathbf{g}_{h,k}(\boldsymbol{\theta}_k)\|^2\right]$ as follows:

$$
\begin{aligned}
\mathbb{E}\left[\|\mathbf{g}_{h,k}(\boldsymbol{\theta}_k)\|^2\right] = \mathbb{E}\left[\|\frac{1}{N}\sum_{i=1}^{N} h_{i,k}\mathbf{g}_{i,k}(\boldsymbol{\theta}_k)\|^2\right] &\leq \frac{N}{N^2}\sum_{i=1}^{N} h_{i,k}^2\|\mathbf{g}_{i,k}(\boldsymbol{\theta}_k)\|^2 \\
&\leq \frac{1}{N}\left(8(\|\boldsymbol{\theta}_k\|^2 + \sigma^2)\sum_{i=1}^{N}\mathbb{E}\left[h_{i,k}^2\right]\right) \leq \left(8(2\delta_k^2 + 3\sigma^2)\right)p_h \\
&= 16p_h\mathbb{E}\left[\delta_k^2\right] + 24p_h\sigma^2.
\end{aligned}
\tag{52}
$$

Hence,

$$
\mathbb{E}\left[\|\mathbf{v}_k\|^2\right] \leq 16p_h\mathbb{E}\left[\delta_k^2\right] + 24p_h\sigma^2 + \frac{\tilde{\sigma}_{\mathbf{w}}^2 d}{N^2}.
\tag{53}
$$

We thus have

$$
\begin{aligned}
\mathbb{E}\left[\delta_{k+1}^2\right] &\leq (1+\alpha)\mathbb{E}\left[\delta_k^2\right] + 2\alpha\left(16p_h\mathbb{E}\left[\delta_k^2\right] + 24p_h\sigma^2 + \frac{\tilde{\sigma}_{\mathbf{w}}^2 d}{N^2}\right) \\
&\leq (1+33p_h\alpha)\mathbb{E}\left[\delta_k^2\right] + 48\alpha p_h\sigma^2 + 2\alpha\frac{\tilde{\sigma}_{\mathbf{w}}^2 d}{N^2}.
\end{aligned}
\tag{54}
$$

Iterating this inequality, we obtain

$$
\mathbb{E}\left[\delta_{2\tau}^2\right] \leq (1+33\alpha p_h)^{2\tau}\delta_0^2 + \left(48\alpha p_h\sigma^2 + 2\alpha\frac{\tilde{\sigma}_{\mathbf{w}}^2 d}{N^2}\right)\sum_{j=0}^{2\tau-1}(1+33\alpha p_h)^j.
\tag{55}
$$

Now with the same procedure used to obtain (32), we see that if $66\alpha p_h\tau \leq \frac{1}{4}$, then

$$
\begin{aligned}
\mathbb{E}\left[\delta_{2\tau}^2\right] &\leq 2\mathbb{E}\left[\delta_0^2\right] + 4\tau\left(48\alpha p_h\sigma^2 + 2\alpha\frac{\tilde{\sigma}_{\mathbf{w}}^2 d}{N^2}\right) \\
&\leq 2\mathbb{E}\left[\delta_0^2\right] + 192\alpha\tau\sigma^2 p_h + \frac{8\alpha\tilde{\sigma}_{\mathbf{w}}^2 d\tau}{N^2} \\
&\leq 2\mathbb{E}\left[\delta_0^2\right] + \sigma^2 + \frac{\tilde{\sigma}_{\mathbf{w}}^2 d}{N^2},
\end{aligned}
\tag{56}
$$

where we have used that $\alpha\tau \leq \frac{1}{878p_h}$. With the choice of step size in the statement of the theorem, we have $\alpha\omega(1-\gamma)m_h \leq 1$. This then yields $(1 - \alpha\omega(1-\gamma)m_h)^{-2\tau} \leq (1 - \alpha m_h)^{-2\tau}$. Finally note that since $\alpha\tau \leq \frac{1}{C_0 p_h} \leq \frac{1}{4p_h}$, we have $\alpha\tau m_h \leq \frac{m_h}{4p_h} \leq \frac{1}{4}$, where we used the fact that $\frac{m_h}{p_h} \leq 1$. Based on this discussion and using Bernoulli's inequality, we obtain $(1 - \alpha m_h)^{2\tau} \geq 1 - 2\alpha\tau m_h \geq \frac{1}{2}$; hence, $(1 - \alpha m_h)^{-2\tau} \leq 2$. We thus have $(1 - \alpha\omega(1-\gamma)m_h)^{-2\tau} \leq 2$. Plugging this bound back in (49) completes the proof.

## APPENDIX B

In this appendix, we provide further simulations to corroborate our theoretical findings and to analyse the performance of `OAC-FedTD` under different configurations. To simulate the Markov chain, we adopt the same configuration as Section 4. In Figure 3, we show the performance of `OAC-FedTD` when the number of states is larger with respect to the experiments shown in the results of Section 4. Specifically, we consider a Markov chain with $|\mathcal{S}| = 100$ states. As expected, the overall number of iterations required for convergence is larger compared to the case in which $|\mathcal{S}| = 20$. However, the linear speedup property of `OAC-FedTD` is evident in this case as well, and the obtained result is similar to the one observed in Figure 2 of Section 4. In Figure 4, we show some further results comparing the performance of `QFedTD` under two different values of the standard deviation of the receiver noise, namely $\tilde{\sigma}_{\mathbf{w}} = 0.2$ and $0.8$. We show the performance for $N = 1$ and $N = 15$ agents, with the step size set to $\alpha = 0.2$. Consistent with the theory, we can see how the convergence ball is inflated by the noise at the receiver. We can note, however, that for $N = 15$, thanks to the speedup guaranteed by cooperation, the noise ball inflation is limited and we still get better performance compared to the single-agent case.

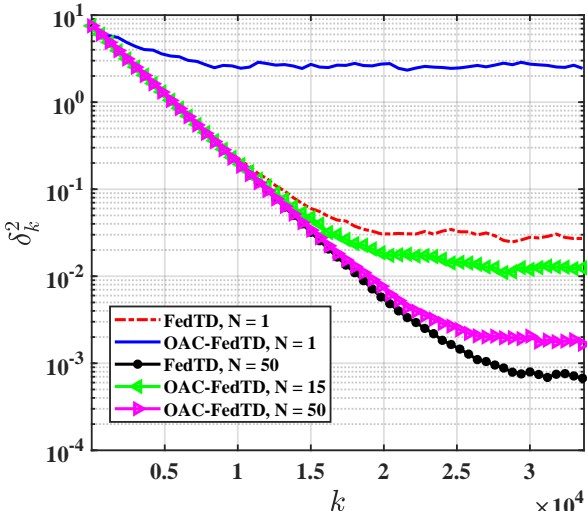

*Figure 3.* Comparison between vanilla `FedTD` and `OAC-FedTD`, in the single-agent ($N = 1$) and multi-agent ($N = 15$, $N = 50$) case. In this experiment, the number of states of the Markov chain is $|\mathcal{S}| = 100$.

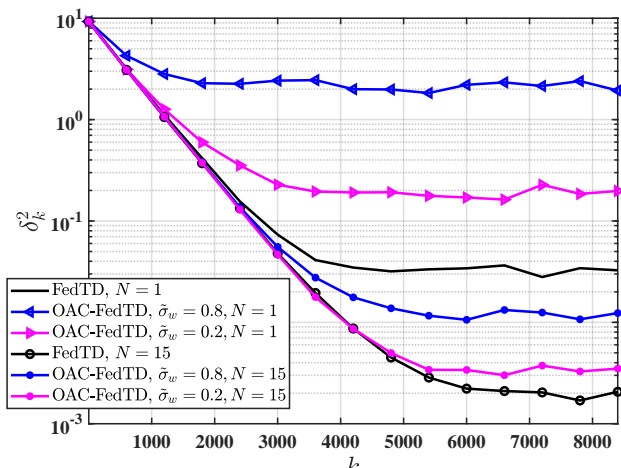

*Figure 4.* Performance of `OAC-FedTD` for different values of the standard deviation of the measurement noise at the receiver ($\tilde{\sigma}_{\mathbf{w}} = 0.2, 0.8$), and for different values of the number of cooperating agents ($N = 1, 15$).