# OpenReview forum: "Over-the-Air Federated TD Learning"
_MLSys/2023/Workshop/RCLWN — MLSys-RCLWN 2023_

### Official Review · Reviewer_B5Ra · 2023-04-29
**Comments on OAC-FEDTD**

**Rating:** 6
**Confidence:** 4

**Review:**

As pointed out in the manuscript, this study is the first work that investigates the influence of noisy fading channels on the convergence rate of federated reinforcement learning (FRL) over wireless fading channels. The combination of over-the-air (OVA) computation and FRL is interesting, and the analysis is solid.

The main concern raised by the reviewer is that the channel model appears to be oversimplified. The reviewer is curious about whether the proposed OAC-FEDTD still performs well if the wireless channel undergoes changes during the training process.

---

### Official Review · Reviewer_rtn1 · 2023-04-30
**This paper is well-written and provides valuable theoretical analysis.**

**Rating:** 10
**Confidence:** 4

**Review:**

This paper studies FRL problems over wireless channels and provides a rigorous finite-time convergence analysis of the OAC-FedTD framework. The paper is well-written, and the theoretical analysis can be a fundamental and useful reference for other FRL-related research works in practical wireless settings. The reviewer has minor following comments:
1. Limited simulation results are included in this paper. If more simulations can be added to this paper, it will be better. For example, the performance with different agent numbers can be analyzed.
2. It should be highlighted how can the theoretical results, such as (5), provide guidance to the practical deployment of FRL algorithms.
3. The authors believe that the channel is still a typically random object even with channel estimation. However, the reviewer does not completely agree with that and still suggests the authors further consider the influence of uplink channel estimation.
4. How do OAC-based FRL algorithms work in MIMO systems? What's the difference in the theoretical analysis?

---

### Official Review · Reviewer_Z82y · 2023-04-30
**This paper studies a federated policy evaluation problem over wireless fading channels, where multiple agents upload local temporal difference (TD) update directions to the central server via over-the-air computation (OAC). The main contribution is to provide a finite-time analysis of the proposed scheme. The theoretical analysis reveals the noisy fading channel effects on the convergence rate and the linear convergence speedup w.r.t. the number of agents of the proposed scheme.**

**Rating:** 7
**Confidence:** 3

**Review:**

### Clarity:
The paper is well-written, and the authors state their central research problem clearly and present their main technical results both theoretically and empirically.

### Originality:
The contributions are novel because there is limited theoretical analysis (as claimed by the authors) pertaining to the convergence behavior of a cooperative reinforcement learning (RL) algorithm under the channel effects introduced by OAC.

### Significance:
The non-asymptotic analysis demonstrates that increasing the number of agents leads to a linear convergence speedup of the RL algorithm and cancels the channel effects, which is theoretically and practically important to reduce the heavy sample-complexity burden of RL.

### Strengths:
- In its presentation, the paper is quite well-written and organized. The authors present quite well the background (based on comprehensive related works analysis) and their propositions. The simulation results are also convincing and validate their theoretical findings.
- The contributions are original and of significance.
- The possible future research directions are also provided.

### Weaknesses:
- The limitations or potential drawbacks of the proposed technique are not discussed.
- More simulation results would be interesting, especially additional results to verify the ***linear*** convergence speedup w.r.t. the number of agents.

---

### Official Review · Reviewer_eGJU · 2023-04-30
**Overall, this paper is well organized and of high quality.**

**Rating:** 9
**Confidence:** 2

**Review:**

In this paper, the authors establish a linear convergence speedup w.r.t. the number of agents for cooperative reinforcement learning (RL) problems subject to realistic communication models and provide a rigorous finite-time convergence analysis, which is novel and can be applied to other federated RL problems. Specifically, they formulate and study federated policy evaluation under the analog OAC model and provide a comprehensive non-asymptotic convergence analysis of OAC-FedTD that simultaneously accounts for Markovian sampling, function approximation, and channel effects.

This paper is fundamental and has rigorous proof. However, a few clarifications are required before possible publication:
1. In Simulation Results, the size n of the state space S is only 20 which is inconsistent with the large state space mentioned in Section Two and the MDP simulation may not be detailed enough.

---

### Meta-Review · Area_Chair_p8RX · 2023-05-11

**Recommendation:** Accept
**Confidence:** 4

**Metareview:**

This paper establishes a non-asymptotic convergence analysis for federated (cooperative) reinforcement learning policy evaluation problem, where wireless channel fading, i.e., Rayleigh fading, is accounted for and analog over-the-air computation is used for uploading each agent's updates to the central aggregator. As pointed out by all four reviewers, and myself agreed, the paper has innovative contributions with solid theoretical results as well as limited yet effective simulation results to validate the proposal. The presentation is clear and easy to follow. That said, major limitations of this work lie in the simulation part, which seems insufficient to fully justify the findigns of the established analysis, in particular, the linear speedup w.r.t. the number of agents. Otherwise, this work is satisfactory. In summary, I recommend accepting this submission and hope to see the authors addressing the concern in simulation results if the space allows.